# The relative impact of injury and deployment on mental and physical quality of life among military service members

Claire Kolaja[1,2]*, Sheila F. Castañeda[1,2], Susan I. Woodruff[3], Rudolph P. Rull[2], Richard F. Armenta[4], for the Millennium Cohort Study Team[¶]

1 Leidos Inc., San Diego, California, United States of America, 2 Deployment Health Research Department, Naval Health Research Center, San Diego, California, United States of America, 3 San Diego State University, School of Social Work, San Diego, California, United States of America, 4 Department of Kinesiology, College of Education, Health and Human Services, California State University, San Marcos, San Marcos, California, United States of America

¶ Membership of the Millennium Cohort Study Team is listed in the Acknowledgments.
* Claire.a.kolaja.ctr@health.mil

**Data Availability Statement:** The data that support the findings of this study are not currently publicly

## Abstract

US service members injured in the recent conflicts in Iraq and Afghanistan were more likely to survive than those in previous conflicts because of advances in medicine and protective gear. However, there is limited research examining the long-term impact of injuries while deployed on physical and mental quality of life (QOL) among service members. We used data from two time-points with an average follow-up period of 4.27 years (SD = 2.13; n = 118,054) to prospectively examine the association between deployment and injury status with QOL. Data were derived from the Millennium Cohort Study surveys (2001 to 2016) and linked with the Department of Defense Trauma Registry (DoD-TR) among a cohort of US service members from all branches and components. The primary predictor (a combination of deployment and injury status) was comprised of the following four categories: 1) not deployed, 2) deployed and not injured, 3) deployed and non-battle injured, and 4) deployed and battle injured. Demographic, military, psychological and behavioral health, and life stress factors were adjusted for in multivariable models. Outcomes of interest were physical and mental QOL from the Short-Form Health Survey for Veterans (VR-36) measured at ~4 year follow-up. Between group comparisons indicated that those deployed and battle-injured had the greatest decline in both mental (-3.82) and physical (-10.13) QOL scores over time (p < .05). While deployment and injury status were associated with poorer mental and physical QOL in adjusted models; only the association between deployment and injury status with physical QOL was clinically meaningful (more than 2.5). In adjusted models, Time 1 mental and physical QOL explained most of the variance (23–25%) in Time 2 mental and physical health QOL as compared to other covariates (e.g., injury and deployment, and other sociodemographic factors increased variance by ~5%). Time 1 QOL was the most significant predictor of later QOL, but those injured while deployed experienced significant and meaningful decrements to long-term physical QOL. This suggests that prevention and rehabilitation interventions should focus on improving physical health among injured service members to avoid long-term adverse effects.

available because of institutional regulations protecting service member survey responses, but they are available on reasonable request and will require data use agreements to be developed. The data use agreement would need to be approved by the NHRC HIPAA Privacy Officer Dr. William Becker and ensure that the dataset is truly de-identified based on Safe Harbor and expert determinations. Requests for data access may be sent to usn.point-loma.navhlthrschcensan.list.nhrc-privacy-list@mail.mil.

**Funding:** The Millennium Cohort Study is funded through the Military Operational Medicine Research Program, Defense Health Program, U.S. Department of Veterans Affairs under work unit no. 60002. The funding agency had no part in the study design, collection of the data, analysis of the data, or writing of manuscript. No financial disclosures were reported by the authors of this paper.

**Competing interests:** Rudolph P. Rull is an employee of the U.S. Government. This work was prepared as part of my official duties. Title 17, U.S. C. §105 provides that copyright protection under this title is not available for any work of the U.S. Government. Title 17, U.S.C. §101 defines a U.S. Government work as work prepared by a military service member or employee of the U.S. Government as part of that person's official duties. Report Number 22-04 was supported by the Military Operational Medicine Research Program, Defense Health Program, and Veterans Affairs under work unit no. 60002. The views expressed in this article are those of the authors and do not necessarily reflect the official policy or position of the Department of the Navy, Department of Defense, nor the U.S. Government. The study protocol was approved by the Naval Health Research Center Institutional Review Board in compliance with all applicable Federal regulations governing the protection of human subjects. Research data were derived from an approved Naval Health Research Center Institutional Review Board protocol, number NHRC.2000.0007.

## Introduction

The survival rate from combat-injuries among military service members deployed in Operation Enduring Freedom/Operation Iraqi Freedom/Operation New Dawn (OEF/OIF/OND) was the highest compared to other conflicts in the past 100 years, primarily due to advanced protective gear and rapid effective medical care [1]. As a result, approximately 56,000 military personnel survived combat-injuries in the OEF/OIF/OND conflicts [2]. Both the causes and effects of combat injuries were distinctive from previous conflicts, with exposures to improvised explosive devices (IEDs) resulting in a high risk of repeated mild and more severe traumatic brain injuries (TBI) [3]. In addition to combat-injuries, approximately a third of injuries sustained in OEF/OIF/OND were non-battle injuries [4]. However, few studies have examined non-battle injuries and even fewer have compared those to battle injuries.

The financial cost of compensating and caring for service members wounded from these conflicts is estimated to exceed $2 trillion [5]. Yet this cost only accounts for diagnosable conditions and benefits, not other subjective consequences that warrant attention. Recent research on the impact of injury and disease has moved beyond the traditional biomedical model, to more ecological models of health and well-being, which recognize that objective disease and disability status alone is insufficient for capturing the long-term individual level impact of illness/injury [6]. Quality of life (QOL) is a concept that, when measured well, can shed light on both the objective and subjective experience of well-being. Typically, QOL is measured through both physical health (i.e., functioning, physical limitations and bodily pain) and mental health (i.e., perceived social functioning, mental health, and general well-being) [7]. QOL measures provide additional contextual and functional information that health care providers may use to have a broader picture of an individual's life circumstances [6], which may be useful when assessing the need for care and rehabilitation [8, 9]. In fact, consideration of mental and physical QOL is now recognized as an integral component of providing healthcare [10].

Research suggests that a range of demographic and psychosocial factors, such as older age, physical health [11, 12], mental health symptoms, unhealthy behaviors (e.g., smoking, poor sleep) [13–16], and life stressors [17] are associated with lower QOL. Additionally, research among service members and veterans has shown that service-related factors, (e.g., deployment, service branch, pay grade, component, and combat injury), pre-existing physical health conditions (e.g., asthma and hypertension), and poor mental health (e.g., PTSD) are adversely associated with QOL [12, 16, 18]. Combat injury, specifically, has been assessed as a QOL correlate to some degree [9, 19] but these studies have methodological limitations (e.g., cross-sectional analyses or convenient sampling), or focused on a specific type of trauma such as limb loss or concussion. A study of health-related QOL among those wounded in combat showed depression and PTSD were associated with lower QOL [18]. Others have reported that deployment, irrespective of combat and whether an injury was acquired, may be associated with decreased psychological health over time [20, 21]. However, there is a dearth of information on how deployment and injury status influence QOL independent of one another. It is important to discern the differential and combined effects of both deployment and injury on health-related QOL to inform future prevention and intervention efforts aimed at improving service member readiness and well-being. Additionally, as non-battle injuries accounted for a significant proportion of injuries sustained while deployed and may increase risk of adverse outcomes, it is also important to include this group in analyses examining deployment and injury on QOL. Therefore, the primary objective of this study was to understand the relative contribution of deployment and injury status (both battle and non-battle) on long-term physical and mental QOL among U.S. service members. We hypothesized that: 1. injured service members will

report poorer QOL than non-injured participants and 2. battle-injured service members will report the overall lowest QOL.

## Materials and methods

### Participants

Data from the Millennium Cohort Study, the largest and longest running cohort study of active duty, Reserve, and National Guard military personnel and veterans, was used for these analyses. Launched in 2001, the Millennium Cohort Study has enrolled over 250,000 participants into five distinct panels between 2001 and 2021 after being randomly selected from active and reserve rosters. All participants provided written informed consent at enrollment and complete comprehensive self-reported surveys every 3 to 5 years that assess factors such as health and behavioral measures, QOL, and military service factors. At enrollment, the active duty participants reflected the composition of the military overall, with most being male (70%), non-Hispanic white (68%), less than 35 years old (83%), of Enlisted pay grade (85%), and serving in the Army (35%) [22]. Participant survey data can be linked to external administrative records to supplement exposure or outcome data. Additional information about study enrollment, methodology, and response rates has been previously published [22, 23]. The study was approved by the Naval Health Research Center IRB (protocol number NHRC.2000.0007).

### Selection procedure

The study sample consisted of 118,054 Millennium Cohort Study participants who met eligibility criteria. For this study, participant records from the first four panels (enrolled from 2001–2013; n = 201,619) were linked to Department of Defense Trauma Registry (DoD-TR) data from January 2002 to July 2016. The DoD-TR is a registry maintained by DoD Center of Excellence for Trauma that collects and maintains records of injuries that occur among deployed service members from the point of injury to final disposition [24]. The registry was initially created to improve battlefield care and assess quality of care. Overall, 1,320 (<1%) Millennium Cohort Study participants were identified in the DoD-TR between 2002 and 2016 of whom 30 were deceased from the injury and two were missing information on injury type (battle versus non-battle) and therefore were excluded.

The main inclusion criterion for this analysis was completion of two Millennium Cohort surveys (hereafter called Time 1 and Time 2). For participants identified in the DoD-TR, the injury date was used to identify the proximal pre- and post- surveys (see Fig 1). Additionally, participants must have completed the QOL measure on Time 1 and Time 2 surveys. Among the 135,854 participants who met the eligibility criteria, 17,800 (13%) were excluded due to missing covariate information at Time 1, resulting in a final study sample of 118,054 participants (see Fig 2).

### Measures

QOL was assessed using the Short-Form Health Survey for Veterans (SF-36V) [25, 26], a QOL measure included on every Millennium Cohort Study survey through the 2014 cycle. The SF-36V is a widely used, standardized QOL measure that relies on participants' self-report of health status and functioning over the past 4 weeks. It contains 36 questions that measure physical and mental health across eight domains: physical functioning, vitality, bodily pain, general health perceptions, physical role functioning, emotional role functioning, social role functioning, and mental health. Domains are combined into physical QOL (physical

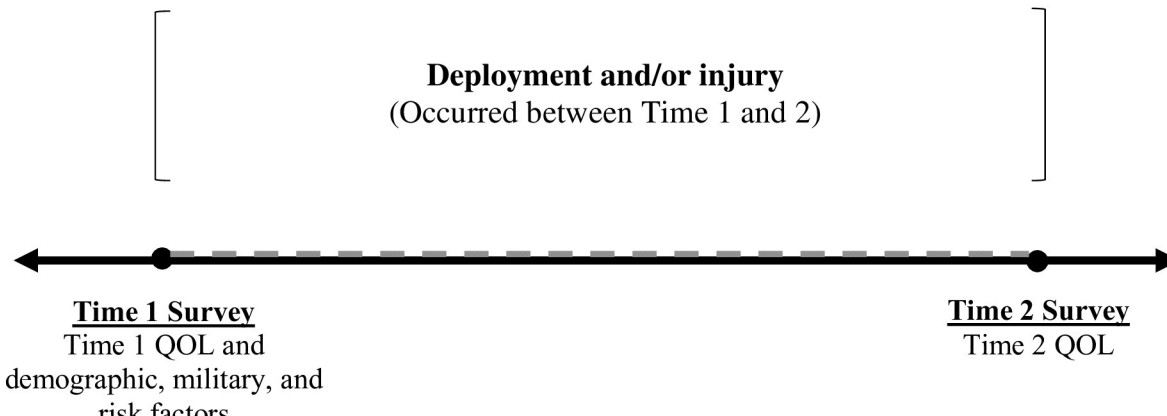

**Fig 1. Timeline for the deployment and injury status on quality of life analysis.**

component summary, or PCS) and mental QOL (mental component summary, or MCS) scales which average the physical and mental relevant questions, respectively. The MCS and PCS scores are computed using standardized methods to compare to US population normative values of 50 (SD = 10) where higher scores indicate better QOL [25, 26]. Please see user's manual for additional details [26].

Using DoD-TR data, Millennium Cohort Study participants were categorized as battle or not-battle injured. Participants not in the DoD-TR were further categorized as deployed and not injured, and not deployed using deployment dates in support of OIE/OIF/OND obtained from the CTS between Time 1 and Time 2. Per nomenclature used in recent articles [27, 28], a four level variable with the following categories was created: (a) deployed and battle injured (DBI), (b) deployed and non-battle injured (DNBI), (c) deployed and not injured (DNI), and (d) not deployed (ND).

All covariates (i.e., age, marital status, education, component, service branch, pay grade, number of deployments, health behaviors, mental health, and life stressors) were assessed at the Time 1 survey. Sex, age, race and ethnicity (i.e., non-Hispanic White, non-Hispanic Black, and other), military component (i.e., active duty, Reserve/National Guard), service branch (i.e., Army, Air Force, Navy/Coast Guard, Marine Corps), and pay grade (i.e., enlisted, officer) were obtained from the Defense Manpower Data Center (DMDC). The Other race and ethnicity category included Hispanic/Latino, Asian American or Pacific Islander, American/Alaskan Indian and other or multiracial (unspecified) service members. Marital status and education were self-reported and backfilled if missing with DMDC data. Dates for deployments in

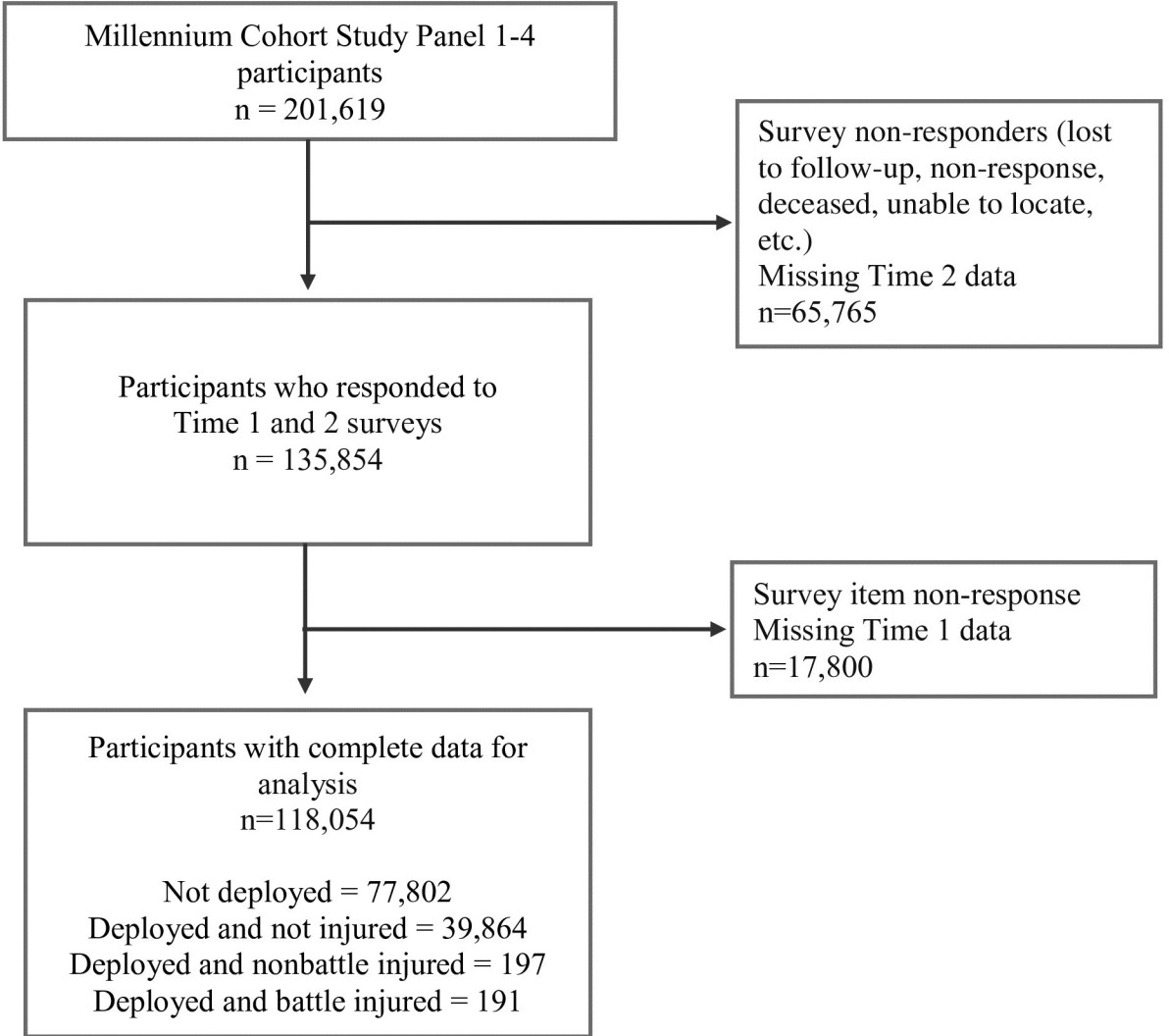

**Fig 2. Flow chart of inclusion criteria for the deployment and injury status on quality of life analysis.**

support of OIE/OIF/OND provided from CTS were used to determine the number of deployments before Time 1.

Based on National Sleep Foundation recommendations sleep duration was categorized as less than 6, 6, 7–9, and more than 9 hours using self-reported average number of hours slept daily in the past month [29]. Smoking status was based on responses to two questions regarding smoking at least 100 cigarettes in their lifetime and cessation such that never smokers smoked less than 100 cigarettes, and those who smoked at least 100 cigarettes were classified as former smokers if they reported quitting successfully, or current smokers, if they had not quit [30]. Depression symptoms (no/yes) were assessed using the PHQ depression scale (PHQ-8) based on DSM-IV-TR criteria. Participants screened positive for depression symptoms based on the following criteria: responded "more than half the days" or "nearly every day" to at least 5 items with at least one item endorsed being depressed mood or anhedonia [31]. PTSD symptoms (no/yes) were assessed using the PTSD Checklist—Civilian Version (PCL-C). The measure consists of 17 items that evaluate PTSD symptoms during the previous month. PTSD

symptoms were determined based using the criteria as defined in the DSM-IV [32] for participants who reported "moderate" or greater on at least one intrusion symptom, three avoidance symptoms, and two hyperarousal symptoms. Chronic life stressors were categorized as none, one, or two or more based on endorsement of the following stressful life events: divorce, financial problems, physical assault, sexual assault, and illness or death of a loved one [33]. Time between the two surveys was calculated in years.

## Statistical analyses

Means and standard deviations (SD) for Time 1 and Time 2 MCS, PCS and change in MCS and PCS were calculated by injury/deployment status (Table 1). The mean differences of Time 1, Time 2, and change over time in MCS and PCS between the four injury/deployment groups were compared using Tukey post hoc tests. Descriptive statistics (frequencies and percentages or means and standard deviation) were calculated for all covariates by the injury/deployment status (Table 1). Collinearity was assessed among the predictors and was tested with a variance inflation factor threshold of $\geq 4$.

Hierarchical linear regression was employed to examine the association between injury/deployment status with MCS and PCS at Time 2. For both QOL outcomes three models were run: Model 1 included Time 1 MCS for the MCS outcome and Time 1 PCS for the PCS outcome; Model 2 added the injury/deployment status to Model 1; Model 3 added time between the surveys, demographic factors (i.e., age, sex, race and ethnicity, marital status, education), military factors (i.e., component, service branch, pay grade and number of deployments), and other factors (i.e., sleep duration, smoking status, major depression, PTSD, life stressors) to Model 2. $R^2$ and root mean square error (MSE) were reported for each model (see Tables 2 and 3 for MCS and PCS, respectively). Statistical analyses were performed using SAS software version 9.4 [34].

## Results

### Descriptive analyses

Among the 118,054 eligible participants, 77,802 ND (65.9%), 39,864 DNI (33.8%), and 388 were injured (0.3%; 197 DNBI and 191 DBI). The sample was 69.3% male, and 75.8% non-Hispanic White with a mean age of 31.9 years at Time 1 (SD = 9.0; Table 1). Most participants were married (59.0%), had less than a college degree (55.8%), served on active duty (60.7%), were enlisted personnel (77.6%), and almost half served in the Army (45.4%). The majority did not screen positive for depression or PTSD symptoms (96.0% and 93.8%, respectively), reported recommended health behaviors (e.g., 47.5% an average of 7–9 hours of sleep, 59.0% never smoked cigarettes) and reported no life stressors (58.7%) at Time 1. Compared to those who were DNI, DBI were more likely to be male, active duty, enlisted pay grade, and serving in the Army. The average time between the Time 1 and Time 2 surveys was 4.3 years (SD = 2.1) and the average time between injury and Time 2 survey was 2.3 (SD = 2.0). Fully adjusted models adjusted for variables listed in Table 1 (see Tables 2 and 3).

### Between group differences in QOL

Mean MCS at Time 1 varied by deployment and injury status (mean of 51.1 for ND, 52.1 for DNI, 52.3 for DNBI, and 50.9 for DBI; p-value < .0001; Table 1 and Fig 3) and pairwise comparisons using Tukey multiple comparison procedures revealed a significant (p-value < 0.05) group difference between ND and DNI's mean scores. Mean change in MCS between Time 1 and Time 2 also varied significantly by deployment and injury status (mean change for ND

**Table 1. Demographic, behavioral, and military characteristics by injury status among Millennium Cohort participants, n = 118,054.**

| Variable | Not deployed (n = 77,802) | Deployed and not injured (n = 39,864) | Deployed and injured Nonbattle (n = 197) | Deployed and injured Battle (n = 191) |
|---|---|---|---|---|
| | N (%) | N (%) | N (%) | N (%) |
| Time between Time 1 and Time 2 surveys, in years (Mean, SD) | 3.72 (1.89) | 5.31 (2.17) | 5.96 (3.29) | 5.46 (3.03) |
| **MCS**[*] | | | | |
| Time 1 (Mean, SD) | 51.10 (10.41) | 52.08 (9.01) | 52.31 (8.85) | 50.93 (10.10) |
| Change score (Mean, SD) | -1.19 (10.45) | -1.96 (10.81) | -3.35 (11.95) | -3.82 (13.20) |
| Time 2 (Mean, SD) | 49.90 (11.24) | 50.12 (10.73) | 48.96 (12.36) | 47.11 (13.00) |
| **PCS**[**] | | | | |
| Time 1 (Mean, SD) | 53.04 (8.06) | 54.89 (6.31) | 54.20 (6.36) | 54.35 (6.68) |
| Change score (Mean, SD) | -0.74 (8.08) | -2.24 (8.14) | -8.80 (11.78) | -10.13 (11.78) |
| Time 2 (Mean, SD) | 52.31 (8.49) | 52.66 (7.95) | 45.40 (11.48) | 44.23 (11.05) |
| **Covariates** | | | | |
| Age, in years (mean, SD) | 32.93 (9.37) | 30.05 (7.76) | 29.19 (7.23) | 28.16 (6.07) |
| Sex | | | | |
| Male | 51,241 (65.86) | 30,199 (75.76) | 162 (82.23) | 178 (93.19) |
| Female | 26,561 (34.41) | 9,665 (24.24) | 35 (17.77) | 13 (6.81) |
| Race and Ethnicity | | | | |
| Non-Hispanic Black | 8,745 (11.24) | 4,063 (10.19) | 27 (13.71) | 17 (8.90) |
| Non-Hispanic White | 58,813 (75.59) | 30,381 (76.21) | 146 (74.11) | 154 (80.63) |
| Other (including Hispanic, Asian/Pacific Islander, American/ Alaskan Indian and other) | 10,244 (13.17) | 5,420 (13.60) | 24 (12.18) | 20 (10.47) |
| Marital Status | | | | |
| Single | 21,002 (26.99) | 13,371 (33.54) | 60 (30.46) | 55 (28.80) |
| Married | 46,931 (60.32) | 22,531 (56.52) | 112 (56.85) | 125 (65.45) |
| Separated | 9,869 (12.68) | 3,962 (9.94) | 25 (12.69) | 11 (5.76) |
| Education | | | | |
| Less than college | 42,086 (54.09) | 23,497 (58.94) | 133 (67.51) | 134 (70.16) |
| College degree | 35,716 (45.91) | 16,367 (41.06) | 64 (32.49) | 57 (29.84) |
| Component | | | | |
| Reserve/ National Guard | 33,231 (42.71) | 13,008 (32.63) | 54 (27.41) | 56 (29.32) |
| Active duty | 44,571 (57.29) | 26,856 (67.37) | 143 (72.59) | 135 (70.68) |
| Service Branch | | | | |
| Army | 34,848 (44.79) | 18,488 (46.38) | 147 (74.62) | 165 (86.39) |
| Navy or Coast Guard | 16,520 (21.23) | 5,223 (13.10) | 13 (6.60) | 3 (1.57) |
| Marine | 6,584 (8.46) | 2,676 (6.71) | 15 (7.61) | 14 (7.33) |
| Air Force | 19,850 (25.51) | 13,477 (33.81) | 22 (11.17) | 9 (4.71) |
| Pay Grade | | | | |
| Enlisted | 60,713 (78.04) | 30,581 (76.71) | 165 (83.76) | 160 (83.77) |
| Officer | 17,089 (21.96) | 9,283 (23.29) | 32 (16.24) | 31 (16.23) |
| Number deployments (Mean, SD) | 0.37 (0.81) | 0.61 (1.19) | 0.45 (0.70) | 0.40 (0.64) |
| Average hours of sleep | | | | |
| Less than 6 hours | 14,206 (18.26) | 7,389 (18.54) | 49 (24.87) | 45 (23.56) |
| 6 hours | 23,810 (30.60) | 12,757 (32.00) | 61 (30.96) | 74 (38.74) |
| 7–9 hours | 37,257 (47.89) | 18,718 (46.95) | 79 (40.10) | 65 (34.03) |
| More than 9 hours | 2,529 (3.25) | 1,000 (2.51) | 8 (4.06) | 7 (3.66) |
| Smoking status | | | | |
| Never | 45,408 (58.36) | 24,064 (60.37) | 107 (54.31) | 94 (49.21) |

*(Continued)*

**Table 1.** (Continued)

| Variable | Not deployed (n = 77,802) | Deployed and not injured (n = 39,864) | Deployed and injured | |
|---|---|---|---|---|
| | | | Nonbattle (n = 197) | Battle (n = 191) |
| | N (%) | N (%) | N (%) | N (%) |
| Former | 19,483 (24.04) | 8,823 (22.13) | 46 (23.35) | 46 (24.08) |
| Current | 12,911 (16.59) | 6,977 (17.50) | 44 (22.34) | 51 (26.70) |
| Depression symptoms (yes) | 3,635 (4.67) | 1,039 (2.61) | 6 (3.05) | 7 (3.66) |
| PTSD symptoms (yes) | 5,580 (7.17) | 1,671 (4.19) | 6 (3.05) | 13 (6.81) |
| Life Stressors | | | | |
| None | 43,291 (55.64) | 25,804 (64.73) | 118 (59.90) | 120 (62.83) |
| One | 21,196 (27.24) | 9,433 (23.66) | 54 (27.41) | 51 (26.70) |
| 2 or more | 13,315 (17.11) | 4,627 (11.61) | 25 (12.69) | 20 (10.47) |

MCS—Mental Composite Summary; PCS—Physical Composite Summary; MCS and PCS are derived from the Short-Form Health Survey for Veterans measure

*Mean MCS at Time 1 varied significantly by deployment and injury status (p-value < .0001). Pair wise comparisons using Tukey multiple comparison procedure revealed one significant (p-value < 0.05) group difference in the mean MCS at Time 1 between ND and DNI. Mean change in MCS between Time 1 and Time 2 varied significantly by injury/deployment status (p-value < .0001). Pair wise comparisons using Tukey multiple comparison procedure revealed two significant (p-value < 0.05) group differences in the change in MCS between 1) DBI and ND, and 2) DBI and DNI. Mean MCS at Time 2 varied significantly by injury/deployment status. Pair wise comparisons using Tukey multiple comparison procedure revealed three significant differences (p-value < 0.05) in mean MCS at Time 2 between 1) ND and DNI, 2) DBI and DNI, and 3) DBI and ND.

**Mean PCS scores at Time 1 varied significantly by deployment and injury status (p-value < .0001). Pair wise comparisons using Tukey multiple comparison procedure revealed one significant (p-value < 0.05) group difference in the mean PCS between those DNI and ND. Mean change in PCS between Time 1 and Time 2 varied significantly by injury/deployment status (p-value < .0001). Pair wise comparisons using Tukey multiple comparison procedure revealed significant (p-value < 0.05) group differences in the change in PCS between all groups except those DNBI and DBI. Mean PCS at Time 2 varied significantly by injury/deployment status. Pair wise comparisons using Tukey multiple comparison procedure revealed significant difference (p-value < 0.05) in mean PCS at Time 2 between all comparisons except between DNBI and DBI.

Smoking status—defined as never and former/current based off of responses to questions regarding smoking at least 100 cigarettes in their lifetime and successfully quitting smoking cigarettes.

Depression symptoms—assessed using the PHQ depression scale (PHQ-8) based on DSM-IV-TR criteria. Participants screened positive for depression based on the following criteria: responded "more than half the days" or "nearly every day" to at least 5 items with at least one item endorsed being depressed mood or anhedonia.

PTSD symptoms—assessed using the PTSD Checklist—Civilian Version. The measure consists of 17 items which evaluate PTSD symptoms during the previous month. A positive PTSD screen was determined based using the criteria as defined in the Diagnostic and Statistical Manual of Mental Disorders, Fourth Edition, Text Revision.

Life stressors—based on the number of endorsed stressful life experiences in the past 3 years including endorsement of divorce, financial problems, physical assault, sexual assault, and illness or death of a loved.

-1.2, DNI -2.0, DNBI -3.3, and DBI -3.8; p-value < .0001; Table 1 and Fig 3). Pairwise comparisons revealed three significant (p-value < 0.05) group differences in the change in MCS between those ND and the three deployed groups. Finally, mean MCS at Time 2 varied significantly by deployment and injury status (Table 1 and Fig 3) and pairwise comparisons revealed three significant differences (p-value < 0.05) in mean MCS at Time 2 between 1) ND and DNI, 2) DNI and DBI, and 3) ND and DBI.

Similar trends were observed for PCS with more pronounced differences observed by deployment and injury status. Mean PCS at Time 1 varied by deployment and injury status (mean of 53.0 for ND, 54.9 for DNI, 54.2 for DNBI, and 54.4 for DBI; p-value < .0001; Table 1 and Fig 4) and pairwise comparisons revealed a significant (p-value < 0.05) group difference between ND and DNI's mean Time 1 PCS. Mean change in PCS between Time 1 and Time 2 varied significantly by deployment and injury status (mean change for ND -0.7, DNI -2.2, DNBI -8.8, and DBI -10.1; p-value < .0001; Table 1 and Fig 4) with significant (p-

**Table 2. Associations between main exposure and covariates with Mental Composite Scores (MCS) among Millennium Cohort participants, n = 118,054.**

| Variable | Model 1 | Model 2 | Model 3 |
|---|---|---|---|
| | Estimate (SE) | Estimate (SE) | Estimate (SE) |
| **Time 1 MCS** | 0.55 (0.00)** | 0.55 (0.00)** | 0.47 (0.00)** |
| **Time 1 PCS** | | | 0.18 (0.00)** |
| **Injury Category (ref: Deployed and not injured)** | | | |
| Deployed and battle injured | | -2.37 (0.70)* | -1.57 (0.68)* |
| Deployed and nonbattle injured | | -1.28 (0.69) | -0.54 (0.67) |
| Not deployed | | 0.33 (0.06)** | 0.10 (0.06) |
| **Time between surveys, in years** | | | -0.24 (0.01)** |
| **Age (5-year change)** | | | 0.54 (0.02)** |
| **Female sex (ref: Male)** | | | -0.50 (0.06)** |
| **Race and Ethnicity (ref: Non-Hispanic White)** | | | |
| Non-Hispanic Black | | | 0.98 (0.09)** |
| Other (including Hispanic, Asian/Pacific Islander, American/Alaskan Indian and other) | | | -0.17 (0.08)* |
| **Marital Status (ref: Single)** | | | |
| Married | | | 0.12 (0.07) |
| Separated | | | 0.23 (0.11)* |
| **College Degree Education (ref: Some college or less)** | | | 0.01 (0.07) |
| **Reserve/National Guard Component (ref: Active Duty)** | | | -0.46 (0.06)** |
| **Service Branch (ref: Army)** | | | |
| Navy/Coast Guard | | | 0.48 (0.08)** |
| Marine | | | -0.39 (0.11)* |
| Air Force | | | 1.63 (0.07)** |
| **Officer Pay Grade (ref: Enlisted)** | | | 0.47 (0.08)** |
| **Number Deployments (cumulative)** | | | -0.51 (0.03)** |
| **Average hours of sleep (ref: 7–9 hours)** | | | |
| < 6 hours | | | -0.86 (0.08)** |
| 6 hours | | | -0.38 (0.06)** |
| 9 < hours | | | -0.63 (0.16)* |
| **Smoking status (ref: never)** | | | |
| Former | | | -0.44 (0.07)** |
| Current | | | -0.62 (0.08)** |
| **Depression symptoms (ref: no)** | | | -0.48 (0.17)* |
| **PTSD symptoms (ref: no)** | | | -2.16 (0.14)** |
| **Life Stressors (ref: none)** | | | |
| One | | | -0.47 (0.07)** |
| 2 or more | | | -1.29 (0.09)** |
| **$R^2$** | 0.248 | 0.248 | 0.294 |
| **Root MSE** | 9.605 | 9.603 | 9.308 |

P-value significance:

*$< 0.05$

**$< 0.0001$

value < 0.05) group differences found between all groups except the two injured groups. Additionally, mean PCS at Time 2 varied significantly by deployment and injury status (Table 1 and Fig 4) where pairwise comparisons again revealed significant difference (p-value < 0.05) between all groups except between the two injured groups.

**Table 3. Associations between main exposure and covariates with Physical Composite Scores (PCS) among Millennium Cohort participants, n = 118,054.**

| Variable | Model 1 | Model 2 | Model 3 |
|---|---|---|---|
| | Estimate (SE) | Estimate (SE) | Estimate (SE) |
| **Time 1 PCS** | 0.53 (0.00)** | 0.53 (0.00)** | 0.48 (0.00)** |
| **Time 1 MCS** | | | 0.08 (0.00)** |
| **Injury Category (ref: Deployed and not injured)** | | | |
| Deployed and battle injured | | -8.14 (0.53)** | -7.72 (0.52)** |
| Deployed and nonbattle injured | | -6.89 (0.52)** | -6.37 (0.51)** |
| Not deployed | | 0.63 (0.05)** | 0.19 (0.05)* |
| **Time between surveys, in years** | | | -0.37 (0.01)** |
| **Age (5-year change)** | | | -0.46 (0.01)** |
| **Female sex (ref: Male)** | | | -0.14 (0.05)* |
| **Race and Ethnicity (ref: Non-Hispanic White)** | | | |
| Non-Hispanic Black | | | -0.31 (0.07)** |
| Other (including Hispanic, Asian/Pacific Islander, American/Alaskan Indian and other) | | | -0.11 (0.06) |
| **Marital Status (ref: Single)** | | | |
| Married | | | -0.37 (0.05)** |
| Separated | | | -0.46 (0.08)** |
| **College Degree Education (ref: Some college or less)** | | | 0.44 (0.05)** |
| **Reserve/National Guard Component (ref: Active Duty)** | | | 0.74 (0.05)** |
| **Service Branch (ref: Army)** | | | |
| Navy/Coast Guard | | | 1.05 (0.06)** |
| Marine | | | 0.46 (0.08)** |
| Air Force | | | 0.63 (0.05)** |
| **Officer Pay Grade (ref: Enlisted)** | | | 0.96 (0.06)** |
| **Number Deployments (cumulative)** | | | -0.37 (0.02)** |
| **Average hours of sleep (ref: 7–9 hours)** | | | |
| < 6 hours | | | -0.95 (0.06)** |
| 6 hours | | | -0.30 (0.05)** |
| 9 < hours | | | -0.11 (0.13) |
| **Smoking status (ref: never)** | | | |
| Former | | | -0.23 (0.05)** |
| Current | | | -0.60 (0.06)** |
| **Depression symptoms (ref: no)** | | | 0.04 (0.13) |
| **PTSD symptoms (ref: no)** | | | -0.32 (0.11)* |
| **Life Stressors (ref: none)** | | | |
| One | | | -0.39 (0.05)** |
| 2 or more | | | -1.03 (0.07)** |
| **$R^2$** | 0.227 | 0.231 | 0.276 |
| **Root MSE** | 7.330 | 7.310 | 7.095 |

P-value significance:

*<0.05

**<0.0001

## Modeling Time 2 Mental Health QOL (MCS)

Table 2 lists results from three hierarchical linear regression models for Time 2 MCS. Model 1, which only included Time 1 MCS, explained almost 25.0% of the variance in Time 2 MCS ($R^2$ = 0.25; root MSE = 9.61). Although the addition of deployment and injury status in Model 2

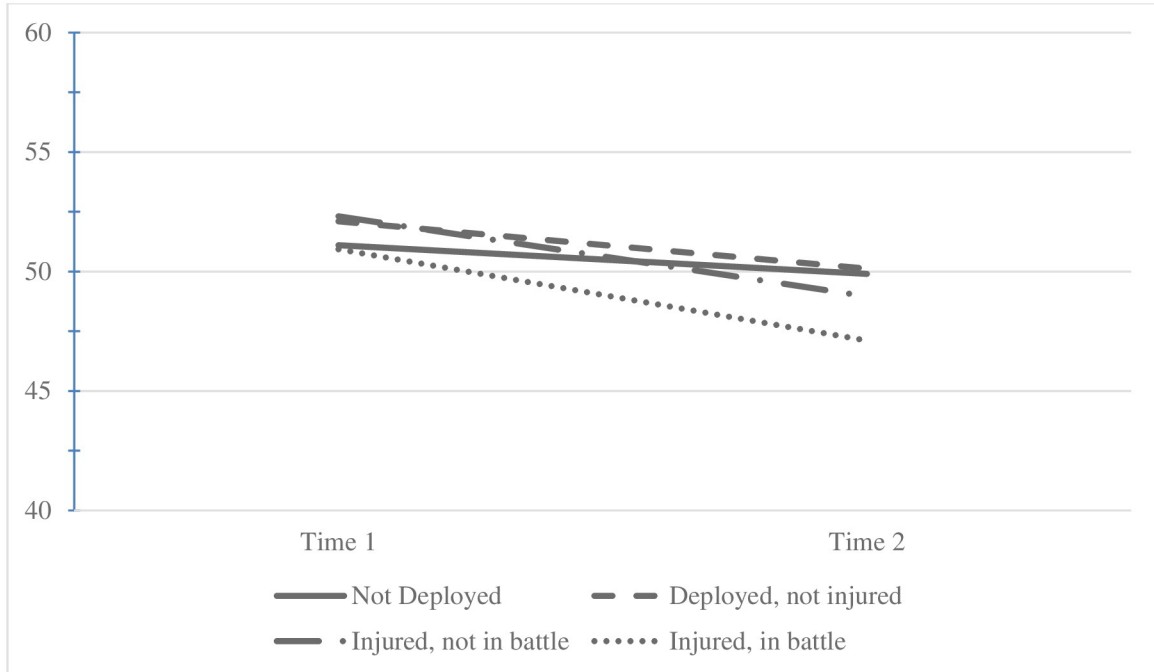

**Fig 3. Average Mental Component Summary by deployment and injury status at Time 1 and Time 2.** Mean MCS at Time 1 varied by deployment and injury status (mean of 51.1 for ND, 52.1 for DNI, 52.3 for DNBI, and 50.9 for DBI; p-value < .0001; Table 1 and Fig 3) and pairwise comparisons using Tukey multiple comparison procedu res revealed a significant (p-value < 0.05) group difference between ND and DNI's mean scores. Mean change in MCS between Time 1and Time 2 also varied significantly by deployment and injury status (mean change for ND -1.2, DNI -2.0, DNBI -3.3, and DBI -3.8; p-value < .0001; Table 1 and Fig 3). Pairwise comparisons revealed three significant (p-value < 0.05) group differences in the change in MCS between those ND and the three deployed groups. Finally, mean MCS at Time 2 varied significantly by deployment and injury status (Table 1 and Fig 3) and pairwise comparisons revealed three significant differences (p-value < 0.05) in mean MCS at Time 2 between 1) ND and DNI, 2) DNI and DBI, and 3) ND and DBL.

did not increase the variance in Time 2 MCS explained, deployment and injury status were significantly associated with Time 2 MCS. Compared to deployed and not injured service members, DBI was associated with lower Time 2 MCS (average of 2.37) and ND was associated with higher Time 2 MCS (average of 0.33). When all covariates were added in Model 3, the variance explained increased by 4.6% ($R^2$ = 0.29; root MSE = 9.31). DBI status was associated with lower Time 2 MCS (average of 1.57). Additional factors associated with lower Time 2 MCS included: time between surveys, female sex, other race and ethnicity, Reserve/National Guard service, Marine Corp (compared to Army), number of deployments, less than 7 or more than 9 average hours of sleep, former or current cigarette smoking, depression symptoms, PTSD symptoms, and life stressors. Conversely, the following characteristics were significantly associated with a higher Time 2 MCS: higher Time 1 MCS and PCS, older age, non-Hispanic Black race and ethnicity, Navy/Coast Guard or Air Force service (compared to Army), and officer pay grade.

### Modeling Time 2 Physical Health QOL (PCS)

The results from the three hierarchical linear regression models for Time 2 PCS are listed in Table 3. Model 1 examined Time 1 PCS on Time 2 PCS and explained almost 23% of the variance in Time 2 PCS ($R^2$ = 0.23; root MSE = 7.33). With the addition of deployment and injury status in Model 2, the variance explained increased by 0.4%. In addition, deployment and injury status was significantly associated with Time 2 PCS, with DBI and DNBI status was

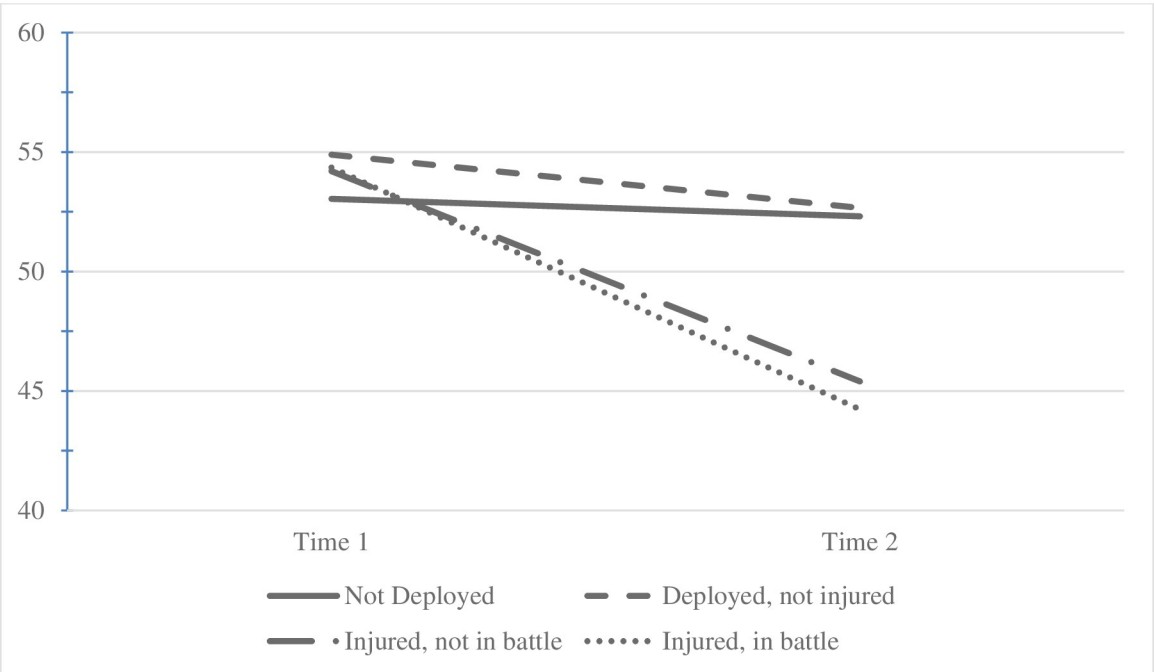

**Fig 4. Average Physical Component Summary by deployment and injury status at Time 1 and Time 2.** Mean PCS at Time 1 varied by deployment and injury status (mean of 53.0 for ND, 54.9 for DNI, 54.2 for DNBI, and 54.4 for DBI; p-value < .0001; Table 1 and Fig 4) and pairwise comparisons revealed a significant (p-value < 0.05) group difference between ND and DNI's mean Time 1 PCS. Mean change in PCS between Time 1 and Time 2 varied significantly by deployment and injury status (mean change for ND -0.7, DNI -2.2, DNBI -8.8, and DBI -10.1; p- value < .0001; Table 1 and Fig 4) with significant (p-value < 0.05) group differences found between all groups except the two injured groups. Additionally, mean PCS at Time 2 varied significantly by deployment and injury status (Table 1 and Fig 4) where pairwise comparisons again revealed significant difference (p-value < 0.05) between all groups except between the two injured groups.

associated with lower Time 2 PCS (average of 8.14 and 6.89, respectively) and ND was associated with higher Time 2 PCS (average of 0.63). Model 3, which included all covariates, additionally increased the variance explained by 4.5% ($R^2$ = 0.28; root MSE = 7.10). In the fully adjusted Model 3, DBI and DNBI were again associated with lower Time 2 PCS (average of 7.72 and 6.37, respectively) and ND was associated with higher Time 2 PCS (average of 0.19). Other characteristics significantly associated with lower Time 2 PCS included: time between surveys, older age, female sex, non-Hispanic Black race and ethnicity, currently married or separated marital status, more deployments, less than 7 average hours of sleep, former or current cigarette smoking, PTSD symptoms, and any life stressors. Factors in Model 3 associated with a higher Time 2 PCS include: higher Time 1 MCS and PCS, college degree, Reserve/National Guard service, Navy or Coast Guard, Marine Corps or Air Force service (compared to Army), and officer pay grade (p-value < 0.001).

## Discussion

This study examined longitudinal changes in mental and physical QOL by deployment and injury status in a large sample of service members. Results showed that although mental and physical QOL decreased for all deployment and injury subgroups, physical QOL significantly and meaningfully decreased among both injured groups, regardless of injury type (i.e., battle or non-battle). The average decreases on the PCS exceed previously identified clinically meaningful difference of 2.5 units on the SF-36 QOL scale [35]. While the between-group difference

on the MCS was statistically significant lower for those injured in battle, the difference did not meet the threshold for clinically meaningful differences (less than 2 points observed difference). As these differences on the PCS are clinically impacted after injury during deployment, regardless of injury type, additional services could be offered to improve physical-related QOL of service members that return from deployment with an injury.

Our analyses showed that deployment and injury status was significantly associated with both mental and physical QOL at Time 2 when adjusting for covariates with the worse QOL observed for those who deployed and were battle injured (DBI). Similar results were observed among a sample of Dutch OEF/OIF veterans (n = 188), where service members with battle casualties reported significantly lower QOL five years later compared to non-injured [36]. However, only an overall QOL score was assessed so direct comparisons cannot be made with mental and physical health constructs. Additionally, one study that compared QOL between battle and non-battle injured Dutch service members also found that those injured in battle experienced lower QOL [37]. Battle injuries that occurred in OEF/OIF/OND were primarily caused by explosive devices [38, 39]; and those DBI (versus DNBI) were more likely to have multiple and severe TBIs [40]. Although not assessed in this study, previous literature has shown that TBIs have a negative effect on long-term quality of life [41].

We also found that those who deployed and had a non-battle injury (DNBI) were more likely to have lower QOL than those who deployed and were not injured (DNI). Few studies have examined QOL after this type of injury [42]. Similar to our findings, a study of Dutch OEF/OIF deployed personnel (n = 223) found significant decreases in physical QOL and no observed difference in mental QOL among those injured with a non-battle injury compared to uninjured deployed service members [28]. This study was a cross-sectional design and possibly unable to assess the long-term impact of injuries. Non-battle injuries among OEF/OIF/OND deployers were primary caused by sports/athletics, fall/jumps, or lifting heavy gear [42, 43]. Although there has been a focus on battle injuries, non-battle injuries account for approximately a third of injuries sustained in OEF/OIF/OND, leading to attrition, increased medical costs, and affecting the readiness of deployed units [4].

In our study, we found that those who did not deploy (ND) had slightly higher physical QOL compared to those who deployed and not injured (DNI), although this was the smallest observed difference (less than 0.20). A similar study that compared personnel deployed to the Gulf War to non-deployers (n = 3,695) found that deployers had significantly lower PCS and MCS scores (-2.1 and -1.9, respectively) [44]. But this study did not separate out those who were injured while deployed, which is a risk factor for poor QOL. Our study examined personnel deployed to the more recent conflicts (OEF/OIF/OND) and found that those not deployed (ND) had similar QOL to those deployed and not injured. Our results also showed a significant difference between mental and physical QOL between ND and DNI at Time 1 (Table 1 and Figs 3 and 4) which corroborates previous research on the Healthy Warrior Effect, suggesting that deployed personnel have optimal health [45–47]. Our results suggest that any advantage deployers had prior to deployment may be attenuated if injured while deployed (either in battle or not).

Contrary to what was hypothesized, most of the variance in Time 2 MCS and PCS was explained by Time 1 MCS and PCS scores. In addition, while having a minimal impact on the total variance for the fully adjusted models, some factors, including deployment and injury status, sleep duration, and smoking status, were found to have a strong and statistically significant association with QOL. Therefore, these factors could be explored in future research and used as potential intervention targets for service members or veterans.

This analysis had notable limitations. Participants were only included if they completed two surveys and survey completers may reflect a unique subset of the original cohort. This may limit the generalizability of the findings to the cohort or the military overall. Also, since the

study used non-injured as the reference groups, we could not account for injury factors such as injury severity, type, or cause. Finally, the PCL-C, which was administered from 2001–2016, was used to assess PTSD symptoms and was based on DSM-IV criteria. Going forward the study will use the PCL-5 which is based on the DSM-V criteria and our researcher has shown that the two measures can equivalently screen for PSTD symptoms in this cohort [48]. Despite these limitations, strengths of the analysis include higher statistical power because of Millennium Cohort Study's large sample size. In addition, inclusion into the Millennium Cohort Study was not conditional on injury status and therefore is more representative of the general military. In addition, the longitudinal design allowed for QOL to be assessed at two time points several years apart and measuring Time 1 covariates prior to the outcomes of interest allows for temporal associations to be examined.

While all groups' QOL worsened over time, certain groups had larger changes. Both constructs of QOL (mental and physical) worsened over time, although the difference in physical QOL reached a clinically meaningful difference for those injured (regardless of whether the injury occurred in battle or not) as compared to those who deployed and were not injured. This indicates that while prior QOL should be considered for research and practice, certain groups, such as those injured while deployed have poorer longer-term physical QOL outcomes, and thus should be considered as potential targets for physical, or vocational rehabilitation interventions.

## Acknowledgments

In addition to the authors, the Millennium Cohort Study team includes Jennifer N. Belding, PhD; Satbir Boparai, MBA; Ania Bukowinski, MPH; Felicia Carey, PhD; Toni Rose Geronimo-Hara, MPH; Clinton Hall, MPH, PhD; Judith Harbertson, MPH, PhD; David Ignacio; Isabel G. Jacobson, MPH; Cynthia A. LeardMann, MPH; Vanessa Perez, MPH; Aprilyn Piega; Anna Rivera, MPH; Rosa Salvatier; Neika Sharifian, MS, PhD; Beverly Sheppard; Steven Speigel; Daniel Trone, PhD; Javier Villalobos, MS; Jennifer Walstrom; and Katie Zhu, MPH. The authors also appreciate contributions from the Deployment Health Research Department, Millennium Cohort Family Study, Birth, and Infant Health Research Team, and Leidos, Inc. We greatly appreciate the contributions of the Millennium Cohort Study participants.

## Author Contributions

**Conceptualization:** Susan I. Woodruff, Richard F. Armenta.

**Data curation:** Claire Kolaja.

**Formal analysis:** Claire Kolaja, Sheila F. Castañeda.

**Investigation:** Claire Kolaja, Sheila F. Castañeda.

**Methodology:** Claire Kolaja, Sheila F. Castañeda, Richard F. Armenta.

**Project administration:** Claire Kolaja, Sheila F. Castañeda, Rudolph P. Rull.

**Software:** Claire Kolaja.

**Visualization:** Claire Kolaja, Sheila F. Castañeda.

**Writing – original draft:** Claire Kolaja, Sheila F. Castañeda, Susan I. Woodruff, Richard F. Armenta.

**Writing – review & editing:** Claire Kolaja, Sheila F. Castañeda, Susan I. Woodruff, Rudolph P. Rull, Richard F. Armenta.

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
