## [Decision Letter · Decision Letter 0]

14 Jun 2022

PONE-D-22-08419The Relative Impact of Injury and Deployment on Mental and Physical Quality of Life among Military Service MembersPLOS ONE

Dear Dr. Kolaja,

Thank you for submitting your manuscript to PLOS ONE. After careful consideration, we feel that it has merit but does not fully meet PLOS ONE’s publication criteria as it currently stands. Therefore, we invite you to submit a revised version of the manuscript that addresses the points raised during the review process.

Please submit your revised manuscript by Jul 29 2022 11:59PM If you will need more time than this to complete your revisions, please reply to this message or contact the journal office at plosone@plos.org. Please include the following items when submitting your revised manuscript:A rebuttal letter that responds to each point raised by the academic editor and reviewer(s). You should upload this letter as a separate file labeled 'Response to Reviewers'.A marked-up copy of your manuscript that highlights changes made to the original version. You should upload this as a separate file labeled 'Revised Manuscript with Track Changes'.An unmarked version of your revised paper without tracked changes. You should upload this as a separate file labeled 'Manuscript'.

We look forward to receiving your revised manuscript.

Kind regards,

Gabriel G. De La Torre

Academic Editor

PLOS ONE

Journal Requirements:

Additional Editor Comments:

Thank you for submitting your manuscript to PLOS ONE. I have now received the required reviews for your manuscript. I review it myself in order to determine whether it (a) is appropriate for this journal (sometimes the topic or focus is more appropriate for another journal; (b), is consistent with current publishing preferences and priorities, or (c) is unlikely to reach the standards usually required for publication. Based on reviewers reports I recommend major revision. I hope that reviewers comments may help you improve the document before resubmitting revision.

Reviewers' comments:

Reviewer's Responses to Questions

**Comments to the Author**

1. Is the manuscript technically sound, and do the data support the conclusions?

Reviewer #1: Partly

Reviewer #2: Yes

2. Has the statistical analysis been performed appropriately and rigorously? 

Reviewer #1: No

Reviewer #2: Yes

3. Have the authors made all data underlying the findings in their manuscript fully available?

Reviewer #1: No

Reviewer #2: Yes

4. Is the manuscript presented in an intelligible fashion and written in standard English?

Reviewer #1: No

Reviewer #2: Yes

5. Review Comments to the Author

Reviewer #1: The introduction requires a better explanation of why this study is necessary. I understand that with such a large sample and a 4-year follow-up this study is quite interesting and relevant. However, the discourse is not well constructed and I did not find it easy to read (I am not a native speaker). The first thing I need to understand is why this study is necessary, what gaps or gaps does it fill that other previous studies have not done, what factors have influenced the quality of life of the participants according to previous studies, have received any kind of treatment in both physical and mental health, which ones? It is also not clear to me what the research problem is. I think that with the potential data and the number of participants the work has to give much more play and statistical analysis based on its longitudinally. It requires figures such as the participant selection procedure and follow-up graphs.

There are many errors when quoting, in the text I have indicated that they should be corrected. For example, instead of "...rapid effective medical care.(1)" it should read "rapid effective medical care (1)." Thus throughout the document.

What were your hypotheses by the way?

The material and methods section should be divided into sections: participants and selection procedure, instruments clearly explained, and data analysis to be performed,...

If you don't explain the instruments well, how can the rest of us interpret the results obtained?

The discussion is not based on the results of the data correctly because they have not done a longitudinal statistical analysis properly.

It is worthwhile for the impact of the work, the number of participants and the collective that they are that this work is polished and better written and presented.

Reviewer #2: This manuscript explores the possible impacts of injury and deployment on the mental and physical quality of life of military service members. The topic, as such, is of great interest to physicians, psychologists, neuroscientists, and other readers, not to mention the military itself. The goal of the study is to understand the relative contribution of deployment and injury status (both battle and non-battle) on the long-term physical and mental Quality of Life of service members. However, this objective strikes me as not clear or concise, i.e., too broad. Establish a quantifiable objective. What are you trying to find out with this study in more operational terms.

In terms of method. Introduce the participants. Show the profile of the participants. Since data on covariates (i.e., age, marital status, education, education, component, service branch, pay grade, 156 number of deployments, health behaviors, mental health, and life stressors) were assessed at the 157 baseline survey. Sex, age at baseline, race and ethnicity (i.e., non-Hispanic White, non-Hispanic 158 Black, and other), present the sample with these data. How old are they? What gender?

Why were PTSD Symptoms described according to the criteria defined in DSM-IV, if DSM V already exists?

Recently, the DSM-5 (American Psychiatric Association, APA, 2013) separated the diagnostic classes for depressive and anxiety disorders. It would be interesting to see if there is an underlying anxiety disorder, independent of PTSD.

What is the future work?

By changing these recommendations I consider that the manuscript is ready for publication.

6. PLOS authors have the option to publish the peer review history of their article (what does this mean?). If published, this will include your full peer review and any attached files.

Reviewer #1: **Yes: **Jose M Mestre, Department of Psychology, Universidad de Cadiz, Spain

Reviewer #2: No

---

## [Author Response · Author response to Decision Letter 0]

5 Aug 2022

Response to Reviewer Comments

PONE-D-22-08419

We appreciate the Reviewers’ thoughtful and positive comments and suggestions. Our responses to Reviewers’ comments and a detailed description of manuscript changes are outlined below in bolded text following the Reviewers’ comments. Our edits to the manuscript are identifiable with highlighted changes as instructed. 

Reviewer #1: The introduction requires a better explanation of why this study is necessary. I understand that with such a large sample and a 4-year follow-up this study is quite interesting and relevant. However, the discourse is not well constructed and I did not find it easy to read (I am not a native speaker). The first thing I need to understand is why this study is necessary, what gaps or gaps does it fill that other previous studies have not done, what factors have influenced the quality of life of the participants according to previous studies, have received any kind of treatment in both physical and mental health, which ones? It is also not clear to me what the research problem is.

RESPONSE: Thank you for pointing out the introduction could be strengthened. To address this, we have modified the introduction for ease of understanding. Specifically, the first paragraph now introduces the prevalence of injured U.S. service members from recent military conflicts; the second paragraph introduces quality of life and the reason to examine it as an outcome; and the third paragraph lays out the previous research on risk factors associated with quality of life, including military specific risk factors. Finally, the end of the introduction now explicitly states the rational for this study.

In the introduction we have highlighted that much of the previous literature has focus on certain type of injuries (such as amputation or concussion) in relation to specific diagnosable outcomes (such as PTSD), lacked a control groups (i.e., those not deployed and those deployed who were not injured), were cross-sectional, or were not able compare battle with non-battled injuries even though non-battle injuries account for a third of causalities. Data from the Millennium Cohort Study was able to address these limitations in a representative, longitudinal cohort where enrollment was not based on injury and was able to examine the combined effect of deployment and injury type on QOL, which has been increasingly recognized as an integral component of healthcare. 

I think that with the potential data and the number of participants the work has to give much more play and statistical analysis based on its longitudinally. 

RESPONSE: We appreciate the reviewer’s concern about the statistical methods used for this project. From the comments we are unsure what statistical analyses are being requested but in reviewing the literature we found that two time point analysis, as we applied, is recommended for longitudinal studies to measure changes in score (Garcia, 2017). This statistical method has advantages as it is simple to apply and understand. 

As the Millennium Cohort Study is an ongoing longitudinal study the analyses conducted for this project relied on survey data available from 2001 to 2016. In addition, this analytic technique has been used for several Millennium Cohort Study projects previously (Adler et al., 2020; Cooper et al, 2020; Jacobson et al., 2021).

References:

Adler, Amy B., et al. "Magnitude of problematic anger and its predictors in the Millennium Cohort." BMC Public Health 20.1 (2020): 1-11. 

Cooper, Adam D., et al. "Mental health, physical health, and health-related behaviors of US Army Special Forces." Plos one 15.6 (2020): e0233560.

Garcia, Tanya P., and Karen Marder. "Statistical approaches to longitudinal data analysis in neurodegenerative diseases: Huntington’s disease as a model." Current neurology and neuroscience reports 17.2 (2017): 1-9.

Jacobson, Isabel G., et al. "Combat Experience, New-Onset Mental Health Conditions, and Posttraumatic Growth in US Service Members." Psychiatry 84.3 (2021): 276-290.

It requires figures such as the participant selection procedure and follow-up graphs.

RESPONSE: Thank you for the suggestion; we have added a new Figure 1 which is a time line and Figure 2 which is a flowchart showing how we arrived at the eligible sample. 

There are many errors when quoting, in the text I have indicated that they should be corrected. For example, instead of "...rapid effective medical care.(1)" it should read "rapid effective medical care (1)." Thus throughout the document.

RESPONSE: Thank you for pointing we were not following the correct formatting guidelines; we have corrected the citations throughout the paper. 

What were your hypotheses by the way?

RESPONSE: We appreciate this suggestion and have added our hypothesis, “1. injured service members will report poorer QOL than non-injured participants and 2. battle-injured service members will report the overall lowest QOL.”, at the end of the introduction. 

The material and methods section should be divided into sections: participants and selection procedure, instruments clearly explained, and data analysis to be performed,...

RESPONSE: To better navigate the Methods Section we have added sub-headers (i.e., “Participants”, “Selection procedure”, and “Measures”) as suggest (see page 5 of the manuscript). 

If you don't explain the instruments well, how can the rest of us interpret the results obtained?

RESPONSE: Thank you for this suggestion, we have provided additional language and references to the methods section to clarifying the instruments used and how they were scored. 

The discussion is not based on the results of the data correctly because they have not done a longitudinal statistical analysis properly.

RESPONSE: As mentioned above, we have applied a recommended and widely used statistical method that has been applied to previous Millennium Cohort Study projects. In this study, participants were measured at two time points, with an average of 4.3 years between the surveys and all risk factors were assessed at baseline (e.g., before the outcomes of interest) so that temporal associations could be examined. 

The discussion section is first divided into paragraphs by the primary predictor, deployment and injury status, then touches on the effect of baseline QOL (see paragraph 5 in the discussion), limitations and strengths of this project, and conclusions. We have edited the discussion throughout to clarifying when we are discussing our findings. 

It is worthwhile for the impact of the work, the number of participants and the collective that they are that this work is polished and better written and presented.

RESPONSE: We appreciate the reviewer’s feedback and suggestions on how to improve the manuscript. The paper has been edited throughout to clarify the rationale for the project and the statistical analyses used. Figure 1 below shows that two Millennium Cohort Study surveys (black points) were used for all eligible participants with an average of 4.3 years between completed surveys. The injured participants completed a Millennium Cohort Study survey before and after the injury, while the rest of the participants were categorized based on deployment date information from administrative data observed between Time 1 and Time 2. We have included a timeline as Figure 1 and a flowchart as Figure 2 to the paper. 

Reviewer #2: This manuscript explores the possible impacts of injury and deployment on the mental and physical quality of life of military service members. The topic, as such, is of great interest to physicians, psychologists, neuroscientists, and other readers, not to mention the military itself. The goal of the study is to understand the relative contribution of deployment and injury status (both battle and non-battle) on the long-term physical and mental Quality of Life of service members. 

However, this objective strikes me as not clear or concise, i.e., too broad. Establish a quantifiable objective. What are you trying to find out with this study in more operational terms.

RESPONSE: Thank you for this suggestion, we have modified the introduction to clarify the aims and added hypotheses at the end of the introduction so that readers understand the rational for this project. 

In terms of method. Introduce the participants. Show the profile of the participants. Since data on covariates (i.e., age, marital status, education, education, component, service branch, pay grade, 156 number of deployments, health behaviors, mental health, and life stressors) were assessed at the 157 baseline survey. Sex, age at baseline, race and ethnicity (i.e., non-Hispanic White, non-Hispanic 158 Black, and other), present the sample with these data. How old are they? What gender?

RESPONSE: We appreciate your suggestions and have add information about the sociodemographic characteristics of the Millennium Cohort Study to the methods section. 

Why were PTSD Symptoms described according to the criteria defined in DSM-IV, if DSM V already exists?

RESPONSE: The Millennium Cohort Study began surveying participants in 2001 before the Diagnostic and Statistical Manual of Mental Disorders updated the case definition for PTSD in 2013. Therefore, this study measured PTSD based on DSM-IV criteria as it was the PTSD measure that was assessed on the surveys between 2001 and 2016. We have added a limitation that the DSM-IV criteria was used for these analyses. The Millennium Cohort Study survey has switched to the PCL-5 in 2019, which corresponds to the DSM-V criteria, so that future studies can utilize that updated version. Additionally, analyses from the 2019 survey cycle found that there is substantial overlap between the two measures (LeardMann et al., 2021).

Reference:

LeardMann, Cynthia A., et al. "Comparison of posttraumatic stress disorder checklist instruments from diagnostic and statistical manual of mental disorders, vs fifth edition in a large cohort of US military service members and veterans." JAMA network open 4.4 (2021): e218072-e218072.

Recently, the DSM-5 (American Psychiatric Association, APA, 2013) separated the diagnostic classes for depressive and anxiety disorders. It would be interesting to see if there is an underlying anxiety disorder, independent of PTSD.

RESPONSE: We appreciate the suggestion to investigate other mental health condition as predictors of QOL. Because of the low prevalence for panic and anxiety, as measured by the Patient Health Questionnaire (approximately 2% and 3%, respectively) and small cell sizes (i.e., less than 10) we were prohibit in using these mental health screeners in these analyses but agree future studies should examine the influence of anxiety symptoms. 

What is the future work?

RESPONSE: We plan on extending our research on the long-term impact of injury and types of injury (e.g., musculoskeletal injuries, traumatic brain injury) to examine other health and economic outcomes among veterans and service members. 

By changing these recommendations, I consider that the manuscript is ready for publication.

RESPONSE: Thank you for taking the time to review this publication.

---

## [Decision Letter · Decision Letter 1]

8 Sep 2022

The Relative Impact of Injury and Deployment on Mental and Physical Quality of Life among Military Service Members

PONE-D-22-08419R1

Dear Dr. Kolaja,

We’re pleased to inform you that your manuscript has been judged scientifically suitable for publication and will be formally accepted for publication once it meets all outstanding technical requirements.

Kind regards,

Darrell Eugene Singer, M.D., M.P.H.

Academic Editor

PLOS ONE

Additional Editor Comments (optional):

Reviewers' comments:

Reviewer's Responses to Questions

**Comments to the Author**

1. If the authors have adequately addressed your comments raised in a previous round of review and you feel that this manuscript is now acceptable for publication, you may indicate that here to bypass the “Comments to the Author” section, enter your conflict of interest statement in the “Confidential to Editor” section, and submit your "Accept" recommendation.

Reviewer #1: All comments have been addressed

Reviewer #2: All comments have been addressed

2. Is the manuscript technically sound, and do the data support the conclusions?

Reviewer #1: Partly

Reviewer #2: Yes

3. Has the statistical analysis been performed appropriately and rigorously? 

Reviewer #1: Yes

Reviewer #2: Yes

4. Have the authors made all data underlying the findings in their manuscript fully available?

Reviewer #1: Yes

Reviewer #2: Yes

5. Is the manuscript presented in an intelligible fashion and written in standard English?

Reviewer #1: Yes

Reviewer #2: Yes

6. Review Comments to the Author

Reviewer #1: They have addressed most of my main concerns. They conducted a large sample and their conclusions were supported by the new amendments. Their article of how prevention and rehabilitation interventions should focus on improving physical health among injured service members to avoid long-term adverse effects might be cited for similar studies with this kind of military samples..

Reviewer #2: Thanks to the authors for taking into consideration the recommendations made by the reviewers.

I value positively the inclusion of the two hypotheses to the work. I would recommend, however, that the hypotheses be quantifiable. What is a "poorer QOL" in operational terms? A hypothesis should be measurable and replicable in future work. This way of describing it can lead to ambiguous interpretation of it.

On the other hand, I appreciate the clarification regarding the participants, but this should be more detailed. So that it helps the reader to have a clear mental scheme of what this sample consists of. There is a lack of data. I think it would help to describe it in more depth (percentage of each sex, percentages of each race and ethnicity, age ranges, etc).

Despite these weaknesses, to be taken into account in future work, I consider the manuscript ready for publication.

Translated with www.DeepL.com/Translator (free version)

7. PLOS authors have the option to publish the peer review history of their article (what does this mean?). If published, this will include your full peer review and any attached files.

Reviewer #1: **Yes: **They have addressed most of my main concerns. They conducted a large sample and their conclusions were supported by the new amendments. Their article of how prevention and rehabilitation interventions should focus on improving physical health among injured service members to avoid long-term adverse effects might be cited for similar studies with this kind of military samples..

Reviewer #2: No

---

## [Editor Report · Acceptance letter]

21 Sep 2022

PONE-D-22-08419R1 

The Relative Impact of Injury and Deployment on Mental and Physical Quality of Life among Military Service Members 

Dear Dr. Kolaja:

I'm pleased to inform you that your manuscript has been deemed suitable for publication in PLOS ONE. Congratulations! Your manuscript is now with our production department. 

Kind regards, 

on behalf of

Dr. Darrell Eugene Singer 

Academic Editor

PLOS ONE